# Regulating Modality Utilization within Multimodal Fusion Networks

**DOI:** 10.3390/s24186054

**Published:** 2024-09-19

**Authors:** Saurav Singh, Eli Saber, Panos P. Markopoulos, Jamison Heard

**Affiliations:** 1Department of Electrical & Microelectronic Engineering, Rochester Institute of Technology, Rochester, NY 14623, USA; esseee@rit.edu (E.S.); jrheee@rit.edu (J.H.); 2Department of Electrical & Computer Engineering and Department of Computer Science, The University of Texas at San Antonio, San Antonio, TX 78249, USA; panagiotis.markopoulos@utsa.edu

**Keywords:** multimodal, data fusion, modality utilization, permutation feature importance, aerial imagery

## Abstract

Multimodal fusion networks play a pivotal role in leveraging diverse sources of information for enhanced machine learning applications in aerial imagery. However, current approaches often suffer from a bias towards certain modalities, diminishing the potential benefits of multimodal data. This paper addresses this issue by proposing a novel modality utilization-based training method for multimodal fusion networks. The method aims to guide the network’s utilization on its input modalities, ensuring a balanced integration of complementary information streams, effectively mitigating the overutilization of dominant modalities. The method is validated on multimodal aerial imagery classification and image segmentation tasks, effectively maintaining modality utilization within ±10% of the user-defined target utilization and demonstrating the versatility and efficacy of the proposed method across various applications. Furthermore, the study explores the robustness of the fusion networks against noise in input modalities, a crucial aspect in real-world scenarios. The method showcases better noise robustness by maintaining performance amidst environmental changes affecting different aerial imagery sensing modalities. The network trained with 75.0% EO utilization achieves significantly better accuracy (81.4%) in noisy conditions (noise variance = 0.12) compared to traditional training methods with 99.59% EO utilization (73.7%). Additionally, it maintains an average accuracy of 85.0% across different noise levels, outperforming the traditional method’s average accuracy of 81.9%. Overall, the proposed approach presents a significant step towards harnessing the full potential of multimodal data fusion in diverse machine learning applications such as robotics, healthcare, satellite imagery, and defense applications.

## 1. Introduction

Continued advancements in technology, data availability, and algorithmic innovation are set to propel the ongoing rise of machine learning. Utilizing statistics to identify and exploit patterns in data is the essence of machine learning. The amount of information in data has a huge impact on how well the machine learning algorithm learns these patterns. Data are also not limited to a single stream of information such as images, audio signals, or text. Multimodal data provide complementary information about the same phenomenon through multiple modalities or information streams. Information captured through multiple modalities can be beneficial in fields such as autonomous driving for image segmentation and object recognition [1], aerospace for activity recognition from aerial imagery [2], robotics for SLAM [3], human–robot collaboration [4], healthcare for diagnosing diseases from medical imagery [5], and defense applications [6]. Certain scenarios may favor one modality over others. For instance, in dark conditions, a near-infrared camera image is more useful for autonomous driving cars than an RGB camera image.

The information from different modalities needs to be merged or fused for a multimodal fusion network to utilize the information from all the different modalities efficiently. Applications within aerial imagery, such as video surveillance, meteorological analysis, vehicle navigation, land segmentation, and activity detection, heavily rely on a diverse array of data sources [7,8,9,10]. These sources encompass various modalities such as electro-optical imaging, synthetic aperture radar (SAR), hyperspectral imaging, and more, each offering unique perspectives and advantages depending on environmental conditions and observational requirements. The data fusion can take place at the data level, network level, and/or decision level [11]. This study focuses on network-level fusion for multimodal remote sensing data, allowing the network to make decisions using data from diverse modalities while learning feature embeddings independently for each modality. This enhances decision-making by exploiting complementary information effectively, resulting in improved accuracy and robustness.

Network-level fusion-based multimodal networks trained end-to-end may inadvertently prioritize one modality over others, resulting in the heavy utilization of a single modality, often neglecting the complementary information offered by others [12,13]. This limits the effectiveness of the multimodal system, especially in scenarios where all modalities contribute equally, or where certain modalities are crucial for accurate inference. During adverse weather conditions like heavy cloud cover, for instance, Electro-Optical imagery might be less effective, while synthetic aperture radar (SAR) could provide clearer insights due to its ability to penetrate clouds. Neglecting SAR data in such conditions could lead to incomplete or inaccurate assessments, highlighting the necessity of balanced modality utilization in aerial imagery applications to ensure robust and reliable outcomes. Furthermore, optimizing multimodal networks in aerial imagery requires the meticulous tuning of hyperparameters tailored to each modality. Variations in learning rates, regularization strengths, and network architectures for different modalities are essential for achieving optimal fusion outcomes [12]. The improper adjustment of hyperparameters for each modality can lead to suboptimal fusion, even causing multimodal networks to underperform compared to their unimodal counterparts.

These issues can be identified by observing the extent to which the network utilizes each of the input data modalities using a modality utilization metric [14]. Recognizing these issues and addressing them is crucial to leverage the multimodal system effectively. This work presents a modality utilization-based training method that can regulate a multimodal network’s utilization of its input data modalities. This helps alleviate the problem of the overutilization of a singular dominant modality. The first research question investigated in this study is, can we leverage the modality utilization metrics during training to regulate a network’s reliance on a dominant modality? The method is validated on a multimodal aerial imagery classification task [15] and a multimodal image segmentation task [16], showcasing its versatility in various multimodal applications. Furthermore, the study investigates the impact of modality utilization-based training on enhancing network robustness to noise, particularly relevant in aerial imagery where environmental factors like weather conditions and time of day can affect different sensing modalities differently. The second research question investigated in this study is, does the modality utilization-based training method improve the overall noise robustness of multimodal fusion networks? The key contributions of this research are as follows:A modality utilization-based training method for multimodal fusion networks to regulate the network’s modality utilization;Demonstrated that regulating modality utilization within a network improves overall noise robustness;A heuristic approach for selecting target utilization-based on unimodal network performance.

The rest of the paper is organized as follows: Section 2 presents related work in the field of multimodal data fusion. Section 3 reviews the modality utilization metric, and presents the modality utilization-based training method. Section 4 lays down the details of the datasets and network architectures used to validate the presented approach. Section 5 presents the experimental results followed by a discussion in Section 6. Section 7 summarizes the findings and future work.

## 2. Related Work

Data fusion in multimodal systems can take place at three different levels: (i) early fusion or data-level fusion; (ii) intermediate fusion or network-level fusion; and (iii) late fusion or decision-level fusion [11]. Data-level fusion consists of combining information or features from different modalities at the raw input level to obtain a better representation of the data prior to the machine learning model [17,18]. Network-level fusion combines information from different modalities within the machine learning model via various mechanisms [19]. Concatenating high-level features or feature embeddings from different modalities within the model is one of the most common methods of network-level fusion. Decision-level fusion takes place at the decision level, where the decisions from multiple unimodal machine learning models are fused into a common decision [19,20]. This work focuses on network-level fusion because it can effectively exploit complementary information from different modalities, learning a joint representation of the multimodal data while being flexible enough to incorporate modality-specific features.

Object detection [21,22,23], image segmentation [24,25,26], and classification [27,28,29] are some of the common tasks in the field of aerial imagery. These tasks heavily rely on multimodal approaches, focusing on applications such as video surveillance, vehicle navigation, land segmentation, and activity detection. These applications use multiple heterogeneous image sources containing inter-modality and cross-modality information such as LiDAR, electro-optical imaging, synthetic aperture radar, hyperspectral imaging, and near-infrared imaging, which need to be combined to enhance the overall information about the phenomenon under observation [30]. Many advances have been made in the field of aerial imagery in recent years, promoted by various global contests such as the IEEE GRSS Data Fusion Contests [8], SpaceNet Challenges [9], and NTIRE challenges [10]. A multimodal knowledge distillation method was proposed by Z. Huang et al. [31] to develop a lightweight CNN model for arbitrary-oriented object detection. This contributes towards the deployment of lightweight models for remote sensing where computational resources are limited. Addressing the limited availability of paired multimodal data, S. Singh et al. [32] used a two-stage training approach for limited multimodal data fusion. Y. Xiang et al. [33] used edge-guided multimodal transformers to detect changes while monitoring land during natural disasters or cloud/fog occlusions based on heterogeneous satellite and aerial image modalities.

Network-level fusion in multimodal networks has access to more information than their unimodal counterparts. Multimodal fusion networks, trained end-to-end, tend to have an imbalance in the utilization of their input modalities due to their greedy nature [34,35]. This bias often results in the overutilization of a single modality, leading to the neglect of the valuable complementary information provided by other modalities [12,13]. Imbalance in multimodal systems limits their effectiveness, especially when all the modalities contribute equally or are crucial for accurate inference. Each modality branch in a multimodal network may require different hyperparameters (e.g., learning rates and regularization strengths) for optimal tuning. Failure to adjust these parameters can lead to suboptimal fusion, causing multimodal networks to underperform compared to their unimodal counterparts. Thus, addressing these challenges is crucial for realizing the full potential of network-level fusion in multimodal systems. M. Ghahremani and C. Wachinger [36] proposed multimodal batch normalization with regularization (RegBN) to tackle the bias and variance issues introduced by heterogeneous modalities. Similarly, I. Gat et al. [37] introduced a regularization term based on functional entropy to address this problem. H. Ma et al. [38] also approached the issue by introducing a regularization term to calibrate predictive confidence when one or more modalities are missing. Relying on multiple modalities also increases the computational complexity of the network. Y. Cao et al. [39] reduced the model complexity of using multiple modalities by learning the common information among the modalities via channel switching and spatial attention. Redundant modalities or modalities with low utility can also be removed based on the learning utility of each modality as demonstrated by Y. He et al. [40]. Another approach to tackle the hyperparameter mismatching across different modalities is to balance the learning of the various modality branches based on an adaptive tracking factor [41]. N. Wu et al. [12] used a conditional utilization rate for each modality based on unimodal performance to tackle the hyperparameter mismatch across different modalities; however, the conditional utilization rate cannot be computed during training time. Since the modality utilization metric can be effectively computed during training [14], the work presented in this paper leverages the modality utilization metric to balance the utilization of the input modalities.

Another rapidly growing approach in the field of multimodal data fusion is transformer architectures, which excel at capturing intricate relationships and long-range dependencies within data, making them suitable for multimodal applications [42]. These architectures are adept at learning the complex interactions among different modalities. Transformers can selectively focus on relevant parts of the input data from each modality by utilizing attention mechanisms [43], thus extracting the most informative features from every source [44]. This allows for more effective fusion of multimodal information, leading to enhanced performance in tasks such as image classification, segmentation, and sequence modeling across modalities. A multimodal fusion transformer (MFT) network featuring a multihead cross-patch attention mechanism was proposed by S. K. Roy et al. [45] for hyperspectral image-based land-cover classification augmented with data from other modalities such as LiDAR and synthetic aperture radar (SAR). Transformers have also been used with heterogeneous modalities such as audio–visual modalities for emotion recognition [46]. Using a CNN encoder and transformer decoder for feature extraction, L. Boussioux et al. [47] developed a tropical cyclone tracking and intensity estimation system using visual data from a reanalysis dataset and historical statistical data. Y. Luo et al. [48] used mixed-attention operations to utilize relatively dominant modality RGB-Thermal tracking in varying environmental conditions to achieve more robust tracking performance compared to single modality tracking systems. This shows that attention mechanisms can enable modality-specific processing, allowing the model to assign varying importance levels to different modalities based on task relevance.

Despite the advancements in multimodal transformer models, visual transformers suitable for aerial imagery require significantly more data compared to their CNN counterparts [49,50]. This poses a limitation on the applicability of transformers for smaller datasets. This study addresses this challenge by introducing a modality utilization-based training method for traditional multimodal neural network architectures. This method aims to regulate the utilization of various input modalities during training, effectively addressing issues such as modality utilization imbalance and hyperparameter mismatch between modalities. Moreover, the field of multimodal aerial imagery suffers from modality utilization imbalance [14,35,51], which is addressed in this work.

## 3. Method

The aim of the proposed method is to regulate a multimodal network’s utilization of its input modalities. This can be achieved by measuring the current utilization of each modality and minimizing the difference between the current and a target utilization for a modality.

### 3.1. Modality Utilization Metric

The modality utilization (MU) metric, proposed in [14], provides a method to quantify a network’s utilization of each modality. The MU metric was inspired by the permutation feature importance [52,53]. Breaking the association between an input modality and the network’s output, the modality utilization is calculated by observing the discrepancies in the multimodal fusion network output when compared to the original dataset.

Assume that there is a dataset D with *M* modalities and *N* samples, and a trained multimodal network $F_{\theta }$ trained on this dataset. Figure 1 shows an example of the dataset and multimodal network with 4 modalities. A forward pass with dataset D on the network $F_{\theta }$ will return the loss (*L*), which is the expected loss L of the network $F_{\theta }$:(1)$$L=E(L\left\{F_{\theta },D\right\})$$

The modality utilization score for the $i^{th}$ modality ($Score_{i}$) is computed by breaking the association between the input modality $M_{i}$ and the network output *Y*. This is performed by randomly shuffling the samples of the modality $M_{i}$ within the dataset D while keeping the samples of the remaining modalities ($M_{j}$ where j≠i) unchanged. A forward pass with the modified dataset $D_{i}$ on the network $F_{\theta }$ will return a new loss ($L_{i}$) of the network $F_{\theta }$:(2)$$L_{i}=E\left(L\left\{F_{\theta },D_{i}\right\}\right)$$

The modality utilization score for the $i^{th}$ modality can then be calculated by observing the discrepancy between the loss *L* and the new loss $L_{i}$:(3)$$Score_{i}=\left|L-L_{i}\right|$$

A larger MU score implies the network performance changed significantly due to changes in the modality $M_{i}$, indicating that the network heavily relies on this modality. A smaller MU score implies that the network performance did not change significantly due to changes in the modality $M_{i}$, suggesting that the network has low utilization of this modality. The MU score can be calculated for each modality and normalized to obtain the modality utilization metric for each modality. The modality utilization $MU_{i}$ ranges from 0.0 to 1.0, providing a percentage utilization of the network $F_{\theta }$ on the $i^{th}$ modality:(4)$$MU_{i}=\frac{Score_{i}}{\sum_{j=1}^{M}Score_{j}}=\frac{|L-L_{i}|}{\sum_{j=1}^{M}\left|L-L_{j}\right|}$$

Algorithm 1 summarizes the modality utilization computation process. The MU metric computation is further explained in more detail with validation and ablation studies in [14]. This metric has also been extended to the reinforcement learning domain, providing valuable insights into the reinforcement learning agents’ action policies [54].
**Algorithm 1:** Modality UtilizationInitialize the multimodal fusion network $F_{\theta }$, learned model parameters θ, task dataset D;Compute the network loss *L* via forward pass with dataset D, Equation (1);
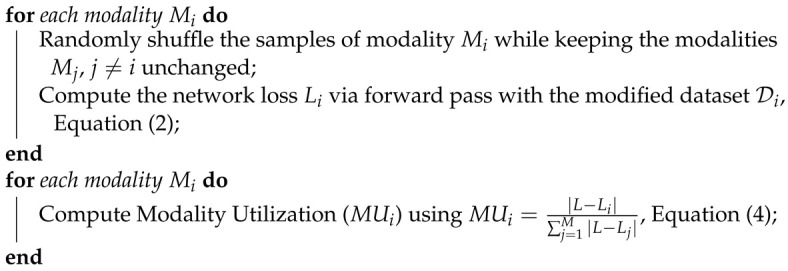


### 3.2. Modality Utilization-Based Training

The modality utilization metric can be leveraged to regulate a multimodal fusion network’s utilization of certain modalities. A network is trained by minimizing a loss function that encapsulates how well the network is performing on a given task, such as classification, segmentation, object detection, and regression. This loss function can be augmented with a second term that minimizes the mean squared error of the current modality utilization ($MU_{curr}$) of the focus modality and the set target modality utilization ($MU_{targ}$). For a generic multimodal fusion network task, the loss function is as follows:(5)$$L_{Total}=L_{task}+\lambda_{L}*L_{mu}$$
(6)$$L_{Total}=L_{task}+\lambda_{L}*MSE\left(MU_{curr},MU_{targ}\right)$$
where $L_{task}$ is the task loss, $L_{mu}$ is the modality utilization loss, and $\lambda_{L}$ is a scaling loss factor. Higher $\lambda_{L}$ emphasizes maintaining target modality utilization for a focus modality, while lower values prioritize solving the fusion network task. Extremely high $\lambda_{L}$ may overemphasize maintaining modality utilization at the expense of solving the fusion task, while $\lambda_{L}=0$ trains the multimodal fusion network conventionally, without modality utilization-based training.

The modality utilization-based training method targets the decision layers of the fusion network to regulate the utilization of the fusion network on its input modalities. Thus, pre-trained frozen weights from the unimodal models are used as feature extractors $FE_{i}\left(M_{i}\right)$ (highlighted in red in Figure 2), and only the decision layers DL are trained (highlighted in green in Figure 2). This approach ensures that the quality of the feature embeddings from each modality is not affected by the modified loss function while allowing for a change in network utilization of its various input modalities.

Algorithm 2 summarizes the modality utilization-based training for the multimodal fusion network. The proposed method compares the current modality utilization of the focus modality with the target modality utilization and optimizes the network to maintain the target utilization while solving the fusion network task.
**Algorithm 2:** Modality utilization-based trainingInitialize the multimodal fusion network $F_{\theta }$, pre-trained feature extractors ($FE_{i}$), task dataset D, focus modality $m_{f}$, loss factor $\lambda_{L}$, and target modality utilization $MU_{targ}$;Load the parameters for pre-trained feature extractors ($FE_{i}$) for each modality;Freeze the pre-trained feature extractors ($FE_{i}$);
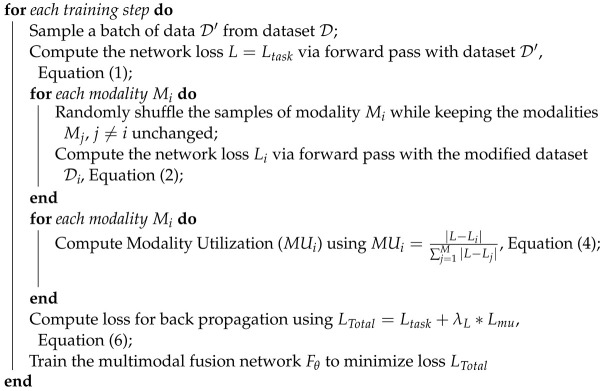


### 3.3. Loss Factor Warm-Up

The proposed training method leverages pre-trained feature extractors to independently extract feature embeddings from each modality. These embeddings are then concatenated and utilized to train the decision layers initialized randomly as illustrated in Figure 3. The initialization of the network plays a crucial role in its convergence towards the global minimum. Initially, both the task loss ($L_{task}$) and the modality utilization loss ($L_{mu}$) are expected to be high. This early stage makes the training process susceptible to being pushed in a local minimum due to the high modality utilization loss ($L_{mu}$) compared to the task-specific loss.

This issue can be resolved by adopting a slow start or a warm-up phase for the loss factor. Setting the loss factor to zero in the initial phase of training allows the network to focus solely on the task loss, gradually ramping up the loss factor after a certain number of training steps stabilizes the modality utilization-based training. A clipped exponential function can provide the desired behavior for the loss factor. The loss factor $\lambda_{L}\left(i\right)$ for the $i^{th}$ training step is as follows:(7)$$\lambda_{L}\left(i\right)=max(0,\lambda_{L\_max}\left(1-e^{\beta (\delta -i)}\right)$$
where $\lambda_{L\_max}$ is the maximum value of the loss factor, β is the buildup rate of the loss factor from 0 to $\lambda_{L\_max}$, and δ is the buildup delay when the loss factor starts increasing exponentially from zero. Figure 4 shows a visualization of the exponential-based loss factor function and hyperparameters. An understanding of the complexity of the task and the architecture of the fusion network can guide the selection of the hyperparameter values.

The maximum value of the loss factor $\lambda_{L\_max}$ dictates the amount of emphasis placed on maintaining the target modality utilization. Given that the computed $MU_{curr}$ and $MU_{targ}$ range between 0.0 and 1.0, $L_{mu}$ (the mean squared error between $MU_{curr}$ and $MU_{targ}$) also ranges between 0.0 and 1.0. Therefore, the maximum value of the loss factor $\lambda_{L\_max}$ must be selected such that the scaled modality utilization loss term $\lambda_{L\_max}\times L_{mu}$ is comparable to the range of the loss of the fusion network task $L_{task}$. A much larger value of $\lambda_{L\_max}$ can render $L_{task}$ insignificant, causing the network to fail to converge properly. Conversely, a much smaller value of $\lambda_{L\_max}$ will make the scaled modality utilization loss term $\lambda_{L\_max}\times L_{mu}$ insignificant, essentially resulting in behavior similar to traditional machine learning training methods.

The buildup rate of the loss factor β dictates how quickly the value of $\lambda_{L}\left(i\right)$ climbs from 0 to $\lambda_{L\_max}$. An aggressive value of β=0.5 is generally a good starting point, as minimal differences in performance are observed with different values of β.

The buildup delay δ is the training step at which the loss factor starts increasing exponentially from 0 to $\lambda_{L\_max}$. The ideal value of δ heavily depends on the complexity of the task and the architecture of the fusion network. The fusion network’s decision layers DL must be trained sufficiently to avoid falling into a local minimum before targeting a modality utilization for the focus modality.

### 3.4. Research Questions

The main research questions for this study are as follows:**RQ1:** Can we leverage the modality utilization metrics during training to regulate a network’s reliance on a dominant modality?**RQ2:** Does the modality utilization-based training method improve the overall noise robustness of multimodal fusion networks?

Two hypotheses are proposed to address these questions:

**Hypothesis** **1** **(H1.)**
*The fusion network, trained with modality utilization-based methods, will effectively maintain the utilization of the focus modality within a margin of error of ±10% in relation to the target utilization.*


**Hypothesis** **2** **(H2.)**
*The noise robustness of a fusion network, trained using a modality utilization-based training method, will vary depending on the target utilization levels of input modalities when noise is introduced to the input modalities.*


## 4. Experimental Design

### 4.1. Datasets and Network Architecture

The modality utilization-based training method for multimodal fusion networks has been validated on a classification task using the NTIRE21 dataset [15]. The presented method is not limited to aerial imagery classification problems and can be employed to different domains and machine learning problems. The versatility of the method is showcased on an image segmentation task using the MCubeS dataset [16].

#### 4.1.1. Classification Task

The NTIRE21 dataset [15] presents a classification problem in the domain of aerial imagery. The dataset consists of aerial views of 10 classes of vehicles: sedan, SUV, pickup truck, van, box truck, motorcycle, flatbed truck, bus, pickup truck with trailer, and flatbed truck with trailer. The multimodal dataset features two modalities: (i) Electro-Optical imagery (EO), and (ii) synthetic aperture radar imagery (SAR). Electro-Optical imagery (EO) is a still image photographic sensing, where the incoming light is converted into electrical signals. Synthetic aperture radar (SAR) is an active imaging method, where the sensor uses microwave radar signals emitted towards Earth to capture surface properties. Figure 5 provides a preview of the NTIRE21 dataset.

The EO modality can capture more information during the day with clear skies; however, the data from the SAR modality may be more reliable during cloudy days or at nighttime. Information from different modalities may be more useful in certain scenarios, and the overutilization of a single dominant modality has the potential to drastically degrade the multimodal network’s performance with noisy data.

Unimodal networks for EO and SAR modalities were trained with ResNet18 [55] as the backbone of the network to obtain the pre-trained feature extractors for the multimodal fusion network. The multimodal fusion network (see Figure 6) consisted of two ResNet18-based pre-trained feature extractor branches for each modality, followed by a flattening layer, concatenation, and a fully connected layer with 512 neurons and a softmax activation function for classification. The network was trained over 250 epochs with an Adam optimizer and a learning rate of 0.001. A significant class imbalance exists in the NTIRE21 dataset, with the *Sedan* class consisting of 234,209 samples while the *Bus* class consisting of only 624 samples. To address this, only the first 624 samples from each class were used for this study. Specifically, 524 samples from each class were used for training, while 100 samples were used for testing. Models with the highest classification accuracy and the lowest modality utilization loss ($L_{mu}$) are saved for further analysis. Additionally, early stopping is employed to prevent overfitting. Figure 7 provides an intuitive demonstration of the predicted classification on the NTIRE21 dataset using the Multimodal Aerial View Object Classification Network.

#### 4.1.2. Image Segmentation Task

The MCubeS dataset [16] presents an image segmentation task in the domain of material segmentation in street scenes. The dataset consists of street scenes from a viewpoint on a road, pavement, or sidewalk containing 20 classes of materials: asphalt, concrete, metal, road marking, fabric, glass, plaster, plastic, rubber, sand, gravel, ceramic, cobblestone, brick, grass, wood, leaf, water, human body, and sky. The multimodal dataset features four modalities: (i) RGB camera (RGB), (ii) Angle of Polarization (AoLP), (iii) Degree of Polarization (DoLP), and (iv) Near Infrared (NIR). Figure 8 provides a preview of the MCubeS dataset. The RGB modality alone cannot capture the necessary information to identify different materials in an image. Different lighting conditions may make certain materials indistinguishable. Thus, other modalities aid in enhancing the overall performance of the multimodal image segmentation network.

Unimodal networks for the RGB, AoLP, DoLP, and NIR modalities were trained with U-Net [56] as the backbone of the network to obtain the pre-trained feature extractors for the multimodal fusion network. The multimodal fusion network (see Figure 9) consisted of four UNet-based pre-trained feature extractor branches for each modality, followed by a concatenation layer, two batch normalization and 2D convolution layers alternately, and a ReLU activation function for image segmentation. The two 2D convolution layers comprised a 3 × 3 × 300 and 1 × 1 × 20 filter size, respectively, with a stride of 1. Training lasted 500 epochs with SGD optimizer (lr: 0.05, momentum: 0.9) on a dataset split into 302 training, 96 validation, and 102 testing samples. Models with the highest mean Intersection over Union (mIoU) and the lowest modality utilization loss ($L_{mu}$) are saved for further analysis. Additionally, early stopping is employed to prevent overfitting. Figure 10 provides an intuitive demonstration of the image segmentation on the MCubeS dataset using the Multimodal Material Segmentation Network.

## 5. Results

### 5.1. Ablation Studies

The presented modality utilization-based training method is first validated on the NTIRE21 dataset and MCubeS dataset without a loss factor warm-up. All experiments were conducted on Rochester Institute of Technology’s research computing server [57] equipped with an Intel Xeon Gold 6150 CPU running at 2.7 GHz, 32 GB of RAM and NVIDIA P4 and A100 GPU’s for NTIRE21 dataset and MCubeS dataset, respectively. The software environment included Red Hat Enterprise Linux 7 as the operating system, Python 3.8.11, and PyTorch 1.10.1 for the deep learning framework. When trained using the traditional method, the multimodal fusion network in NTIRE21 relies heavily on the EO modality, making it the dominant modality (see Table 1) [14]. The utilization of the fusion network in the MCubeS dataset is more evenly distributed when trained using the traditional training method, with RGB as the most utilized modality, while NIR is the least utilized modality. These modality utilization measures are used as the baseline for this study. The NTIRE21 network overemphasizes the EO modality while not utilizing the SAR modality at all. Such a utilization imbalance can be rectified using the presented modality utilization-based training method.

Given a target utilization $MU_{targ}$ and loss factor $\lambda_{L}$, the utilization of a multimodal fusion network on the input focus modality $m_{f}$ can be manipulated using Algorithm 2. The classification network for NTIRE21 dataset is trained with SAR as the focus modality ($m_{f}=SAR$), and loss factor $\lambda_{L}=100$ for different values of target utilization $MU_{targ}$, shown in Figure 11. The network can be observed maintaining the SAR modality utilization $MU_{SAR}$ (green line) close to the target utilization $MU_{targ}$ (black dashed line) while maintaining a performance similar to the baseline methods. The network performance converges to the baseline performance quickly since pre-trained frozen feature extractors are used for this study. Table 2 summarizes the performance and the modality utilization of the network on EO and SAR modalities achieved for different $MU_{targ}$. A drop in performance is observed when the network starts to rely heavily on the non-dominant modality. The performance is bounded by the unimodal SAR performance at $MU_{targ}=100.0\%$. A small difference can be noticed between target modality utilization SAR $MU_{targ}$ and $MU_{SAR}$ as the modality utilization-based training method minimizes the difference between $MU_{targ}$ and $MU_{curr}$ while also solving the fusion network task. A mean difference of 3.73% can be observed between SAR $MU_{targ}$ and $MU_{SAR}$ with a maximum of 7.10% and minimum of 0.59% difference.

Since the measure of MU is a percentage utilization with respect to all the input modalities, the utilization of the EO modality reduces with increasing utilization of the SAR modality. Similar network behavior can be observed in Table 3, where the experiment is repeated with EO as the focus modality ($m_{f}=EO$). A mean difference of 3.76% can be observed between EO $MU_{targ}$ and $MU_{EO}$ with a maximum of 6.90% and minimum of 0.50% difference. The effects of information redundancy in the different modalities can also be observed in Table 2 and Table 3. As the utilization of the dominant modality decreases, the utilization of the non-dominant modality increases; however, the accuracy of the model stays unchanged for $MU_{EO}>50\%$. Redundant information allows for the reduction in the utilization of one modality up to the point where the unique information in that modality begins to be compromised.

The value of the loss factor $\lambda_{L}$ determines the extent to which the MU-based training method emphasizes solving the fusion network task and maintaining the current modality utilization $MU_{curr}$ close to the target utilization $MU_{targ}$. The effects of the loss factor $\lambda_{L}$ on the modality utilization and network performance are demonstrated in Figure 12. The NTIRE21 fusion network was trained with $m_{f}=SAR$, $MU_{targ}=50.0\%$, and $\lambda_{L}=0.0,20.0,100.0,10,000.0$. The MU-based training method works like the traditional machine learning training method with $\lambda_{L}=0$, eliminating the modality utilization loss term from Equation (Equation 5). The NTIRE21 fusion network thus behaves similarly to the baseline network with $\lambda_{L}=0$. As the value of $\lambda_{L}$ is increased, the network maintains the $MU_{curr}$ closer to $MU_{targ}$. However, a really high value of $\lambda_{L}$ can lead to catastrophic failure, as the network only focuses on maintaining modality utilization, completely ignoring the task-specific loss as seen in Figure 12 with an extreme value of $\lambda_{L}=10,000.0$.

The modality utilization-based training method can be applied to a diverse range of multimodal machine learning applications. The method is also validated on the MCubeS image segmentation dataset. The multimodal network is trained for target utilization $MU_{target}$ with a loss factor $\lambda_{L}=100.0$, and RGB (the dominant modality) as the focus modality while observing the mean Intersection over Union (mIoU) and modality utilization of the four input modalities. The results in Table 4 show that the presented method is able to drive the utilization of the network close to the set target for the RGB while maintaining good performance. Since RGB is used as the focus modality, the utilization of the other three modalities is decided by the multimodal network. A mean difference of 4.61% can be observed between RGB $MU_{targ}$ and $MU_{RGB}$ with a maximum of 9.90% and minimum of 1.20% difference. Contrary to modality utilization results from the previous study [14] in Table 1, AoLP appears to be the non-dominant modality instead of the NIR modality with consistent low utilization. Across Table 2, Table 3 and Table 4, a mean difference of 4.03% can be observed between $MU_{targ}$ and $MU_{curr}$ with a maximum of 9.90% and minimum of 0.50% difference.

The results presented in this section were generated over a single fold without modulating $\lambda_{L}$ via loss factor warm-up. As the study is scaled to multifold validation, instability in the modality utilization-based training method can be observed. A five-fold validation for modality utilization-based training methods on the NTIRE21 dataset with $m_{f}=SAR$, $\lambda_{L}=100.0$, and $MU_{targ}=0.0$ reveals that the training method can become unstable with random utilization and data splits, highlighted in red in Table 5.

An untrained network is prone to getting pushed into a local minima by the modality utilization loss term in Equation (Equation 5). The loss factor warm-up presented in Section 3.3 becomes necessary to stabilize the modality utilization-based training method.

### 5.2. Validation Loss Factor Warm-Up

The network parameters with pre-trained feature extractors initially have a smaller task-specific loss, while they may have a larger modality utilization loss. A much higher modality utilization loss in the beginning can push the network away from the minima. The loss factor warm-up allows the network to improve the performance on the network task by keeping $\lambda_{L}=0$ prior to exponentially increasing the $\lambda_{L}$ value. The five-fold study on the NTIRE21 dataset shown in Table 5 is repeated with the loss factor warm-up to validate the loss factor warm start-up. Since the NTIRE21 dataset uses pre-trained frozen data, the network parameters can quickly converge to the global minimum. Thus, a buildup rate of β=0.5 and a buildup delay of δ=0.0 are used, achieving stable performance with consistent results, demonstrated in Table 6.

The buildup delay δ initially suppresses the modality utilization-based training, focusing on the fusion network. Since the feature extractors in the study are pre-trained, a value of δ=0 is used. The buildup rate β dictates how quickly the loss factor $\lambda_{L}$ builds up from 0 to $\lambda_{L\_max}$ in Equation (Equation 7). A large value would aggressively drive $\lambda_{L}$ to the maximum value, making the network focus on maintaining the target utilization early. A smaller value would allow the network to focus on the fusion network task for longer; however, it focuses less on maintaining the target utilization. Figure 13 shows the effects of different buildup rates β on the NTIRE21 classification dataset. A slower building with β=0.01 shows a weaker tendency to maintain the target utilization at SAR $MU_{targ}=50\%$.

### 5.3. Studying Noise Robustness Properties of the Modality Utilization-Based Training Method

The presence of noise in the data can negatively impact the performance of the multimodal network. Noise in one or more modalities may be introduced due to environmental changes or sensor failure. The overutilization of a singular dominant modality makes the multimodal network susceptible to noise in the dominant modality while underutilizing the non-dominant modality. In the case of the NTIRE21 dataset, the EO modality is the dominant modality; however, it is a passive sensing modality that cannot provide usable data during nighttime or on a cloudy day. A more balanced utilization of the input modalities can provide robustness against noise in a singular dominant modality. Modality utilization-based training provides a method to guide the utilization of the multimodal network on its input modalities, improving the network’s overall robustness against noise.

A noise ablation study was conducted with the NTIRE21 dataset by adding Gaussian noise with mean 0 and variance of {0.06,0.09,0.12} to the EO modality, the SAR modality, and both modalities during inference. The effects of noise were studied on networks trained with different levels of SAR utilization, i.e., 0.0%, 12.5%, 25.00%, 37.5%, 50.0%, 62.5%, 75.0%, 87.5%, and 100.0%. The results in Figure 14 show that the overall accuracy is the highest with Clean EO and Clean SAR modalities (indicated by blue) and the lowest with Noisy EO and Noisy SAR (indicated by red). This behavior is expected, as the addition of noise in the data degrades the performance of the network.

When noise is added to only the SAR modality (indicated by green), the performance of models trained with higher SAR utilization is worse than that of the models trained to have higher utilization with the EO modality. Similarly, when noise is added to only the EO modality (indicated by orange), the performance of models trained with higher SAR utilization is better than that of the models trained to have higher utilization with the EO modality. Figure 14 further reveals that network with $MU_{EO}=75\%$ and $MU_{SAR}=25\%$ performs almost the same as the Noisy-EO/Clean-SAR and Clean-EO/Noisy-SAR input modalities. The network exhibits better robustness against noise in either of the modalities compared to networks with other utilizations without significant degradation in performance, indicated by the clean EO and clean SAR performance. This property can be noticed across the different noise levels introduced to the data during inference time.

The fusion network trained with the traditional method self-optimizes to utilize 99.59% of the dominant EO modality and 0.40% of SAR non-dominant SAR modality as indicated in Table 1. Table 7 demonstrates that when noise is present in the dominant modality during inference time, a 75.0% to 87.5% utilization of the EO modality offers better robustness towards noisy dominant modality with minimal loss in accuracy. Compared to traditional training methods that achieve 99.59% EO utilization with 73.7% accuracy under heavy noise conditions (noise variance = 0.12), a network utilizing 75.0% EO performs significantly better with 81.4% accuracy. The network (EO $MU_{targ}=75.0\%$) also achieves an average accuracy of 85.0% across various noise levels, exceeding the traditional method’s average accuracy of 81.9%.

### 5.4. Determining Target Modality Utilization

The target modality utilization $MU_{targ}$ for the focus modality $m_{f}$ is a critical hyperparameter, as it dictates how much the trained network will rely on the focus modality $m_{f}$. Overutilization of the non-dominant modality or underutilization of the dominant modality can lead to network performance degradation. On the other hand, overutilization of the dominant modality or underutilization of the non-dominant modality can affect the robustness to noise in the dominant modality. Thus, choosing an appropriate $MU_{targ}$ is critical; however, it is a non-trivial problem to determine an appropriate target utilization $MU_{targ}$ value without observing the behavior of the network.

Problems such as mismatched hyperparameters across different multimodal branches or overutilization of a single dominant modality are not present in unimodal networks, which are trained on data from a single modality. However, their performance is limited to the information present in those modalities. Thus, unimodal network performance can provide heuristic insights into how well a modality represents information in the context of the fusion network task. Additionally, a multimodal network trained using traditional training methods can be used to determine the multimodal network’s utilization tendencies using the modality utilization metric as seen in Table 1.

Observing unimodal and multimodal network performance that is trained using traditional training methods provides insight into network utilization and information in each modality as viewed by the fusion network. This can guide the selection of the target modality utilization $MU_{targ}$ for the focus modality $m_{f}$ while balancing the utilization of the dominant and the non-dominant modalities. Furthermore, an array of noise robustness tests can reveal certain target utilizations, where the network may be more robust against noise present in one or multiple modalities as shown in Section 5.3. This is crucial for aerial imagery, where varying weather conditions can affect modality usability, especially in datasets lacking diverse weather conditions during training.

## 6. Discussion

The modality utilization-based training method aims to regulate the utilization of multimodal networks on their input modalities. Hypothesis **H1** predicts that the fusion network, trained with modality utilization-based methods, will effectively maintain the utilization of the focus modality within a margin of error of ±10% in relation to the target utilization. In the results in Table 2, Table 3 and Table 4, a mean difference of 4.03% can be observed between $MU_{targ}$ and $MU_{curr}$ with a maximum of 9.90% and minimum of 0.50% difference, thus supporting hypothesis **H1**. The presented method successfully leverages the modality utilization metric to encourage the multimodal network to minimize the mean squared error between the current utilization and maintain a set target for a focus modality. The method alleviates the problem of the overutilization of a singular dominant modality by balancing the utilization among the different input modalities.

However, there needs to be an equilibrium between the network focusing on solving the fusion network task and maintaining the target modality utilization, dictated by a loss factor $\lambda_{L}$. The network optimizes over two different goals, which can drive the network parameters in two different directions. The initial phase of the training process is particularly sensitive, and an instability is observed in the modality utilization-based training method as shown in the results in Table 5. A delayed implementation of the modality utilization-based training method through a warm startup stabilizes the training process, achieving consistent performance.

The aerial imagery classification task with the NTIRE21 dataset presents a case study where the dataset inherently causes a multimodal network to utilize only the dominant EO modality while ignoring the non-dominant SAR modality. The EO modality can provide more information during the daytime with clear sky. However, the SAR modality is more optimal for cloudy weather conditions since the active sensing method can capture information through the clouds. The absence of cloudy data samples leaves out crucial information from the network, driving the network to rely on a singular modality. Expert designers can employ the modality utilization-based training method to train the network correctly even with an imperfect dataset.

The method allows the multimodal network to be more robust towards the presence of noise, such as clouds and other weather conditions, where the EO modality becomes suboptimal. Hypothesis **H2** predicts that the noise robustness of a fusion network, trained using a modality utilization-based training method, will vary depending on the target utilization levels of input modalities when noise is introduced to the input modalities. The results in Figure 14 demonstrate that the presented method can be used to make a network more robust to the presence of noise in one or more modalities by making it less reliant on a dominant modality. The network trained for lower SAR modality utilization performs better than that with higher SAR modality utilization when noise is present in the SAR modality. A similar trend is observed with the EO modality when noise is present in it. This is further validated by Table 7, where a decrease in the utilization of the dominant EO modality decreases the network accuracy when no noise is present. However, the network exhibits a greater degradation in accuracy with higher EO utilization when noise is present in the dominant EO modality. The average optimal EO utilization is revealed to be in the range of 75–87.5%, leveraging the information from the dominant modality while avoiding the overutilization of it. This is an improvement over the traditional training method, which self-optimizes its utilization to be 99.59% utilization of the dominant EO modality, shown in Table 1, making the network susceptible to noise. The array of tests reveals that the network’s performance is degraded by different levels based on the target utilization set for that network, thus supporting hypothesis **H2**. This indicates that there exists an EO-SAR utilization ratio where the network will be equally robust to the presence of noise in the EO or SAR modality, making it the ideal point to improve the overall network noise robustness.

Choosing the optimal target utilization for a modality is critical, as higher utilization of a non-dominant modality can result in poor performance, while overutilization of a singular dominant modality can undermine the information gain from multiple modalities. The unimodal performance of the network for each modality provides heuristic insight into the amount of relevant information in that modality. Furthermore, a multimodal network trained on all the modalities with traditional training methods can reveal how much the network tends to rely on one or the other modality using the modality utilization metric. The modality utilization metric, along with the unimodal performance, offers a holistic understanding of the network and guides the decision of choosing the appropriate target utilization. The aim while choosing the target utilization is to mitigate any overutilization of the network on a single modality, and the target utilization selection process must be motivated by this aim. The target utilization selection can also be made based on the noise robustness properties as demonstrated in Section 5.3; however, this requires conducting an array of tests whose design may be impacted by the number of modalities in the dataset.

The presented method was also validated on an image segmentation dataset in the domain of multimodal material segmentation. This showcases the versatility and applicability of the modality utilization-based training method to a diverse set of multimodal machine learning applications. Furthermore, this method is not limited to multimodal supervised learning applications and can be extended to the domain of reinforcement learning, offering promising avenues for future research and practical implementation in various domains. Overall, this study contributes significantly to advancing multimodal fusion networks, enhancing the utilization of diverse data modalities in machine learning applications.

## 7. Conclusions and Future Work

In conclusion, this work presented a modality utilization-based training method that can be employed to guide the utilization of a multimodal fusion network on its input modalities. The method leverages the modality utilization metric and introduces a modality utilization loss term to minimize the error between the current utilization of a focus modality and a set target. The method was validated on an aerial imagery classification dataset and an image segmentation dataset. The results showed that the presented method can successfully influence a multimodal network’s utilization of its input modalities, effectively maintains modality utilization within ±10% of the user-defined target utilization. Moreover, the network’s robustness against noise in the input modalities was studied, a prevalent challenge in practical scenarios. The method demonstrated higher resilience to input noise affecting different sensing modalities in the NTIRE21 dataset, further enhancing its practical utility. Specifically, networks trained with 75.0% EO utilization exhibited better accuracy (81.4%) under noisy conditions (noise variance = 0.12) compared to traditional methods utilizing 99.59% EO utilization (73.7%). Furthermore, the network maintained an average accuracy of 85.0% across varying noise levels, outperforming the traditional method’s average accuracy of 81.9%. Key contributions include the development of a modality utilization-based training framework, tailored to address utilization imbalances in multimodal fusion networks. The study also offers insights into enhancing network robustness against input noise, advancing the practical utility of multimodal systems.

Future work will validate the approach on networks trained end to end, without relying on pre-trained feature extractors. Additionally, conducting thorough validations on datasets with more than two modalities will enhance the understanding of the interactions between modality utilization and the information content in each modality. Exploring the method’s application with target utilization for multiple modalities and extending it to reinforcement learning are promising avenues for further research. Overall, the findings of this study represent a significant step towards realizing the full potential of multimodal data fusion in machine learning applications, offering promising avenues for future research and practical implementation in diverse domains.

## Figures and Tables

**Figure 1 sensors-24-06054-f001:**
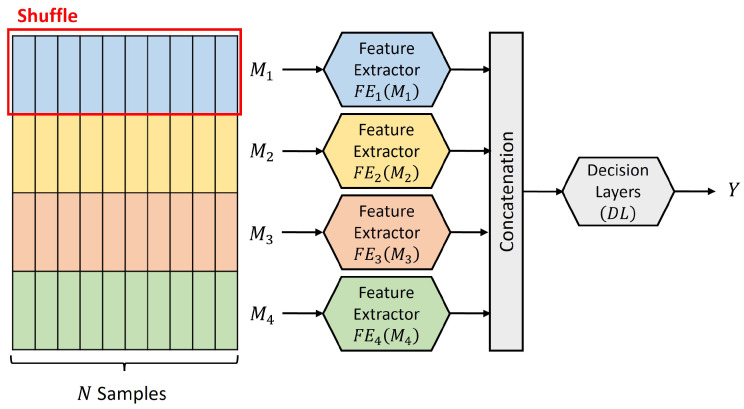
Computing modality utilization by randomly shuffling a modality $M_{i}$ within the dataset to break the association between the input modality $M_{i}$ and the output *Y*.

**Figure 2 sensors-24-06054-f002:**
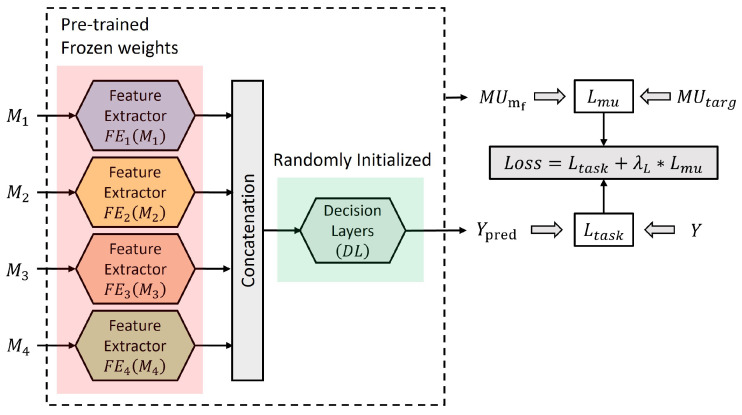
Modality utilization-based training targets the decision layers while using pre-trained feature extractors with frozen weights.

**Figure 3 sensors-24-06054-f003:**
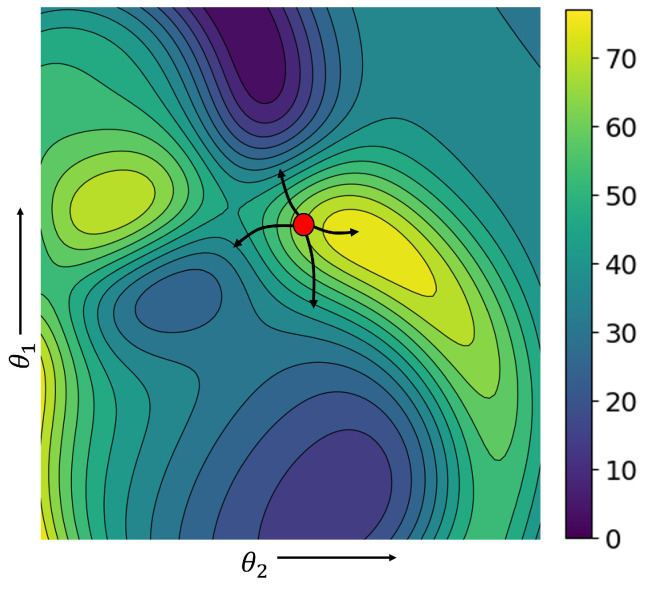
Visualization of multimodal fusion network’s gradient descent on the loss surface of the fusion network task. Optimizing $L_{mu}$ from the very beginning can push the network in the local minima.

**Figure 4 sensors-24-06054-f004:**
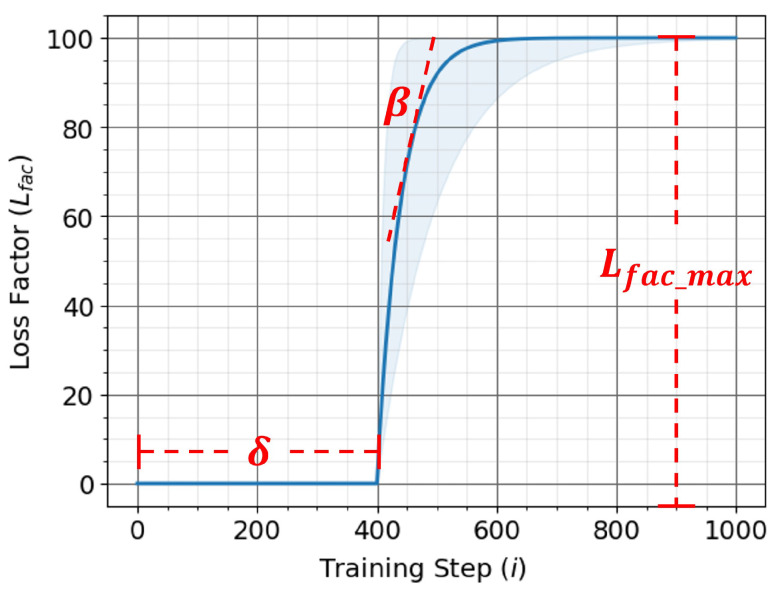
Clipped exponential function-based for loss factor warm-up for MU-based training.

**Figure 5 sensors-24-06054-f005:**
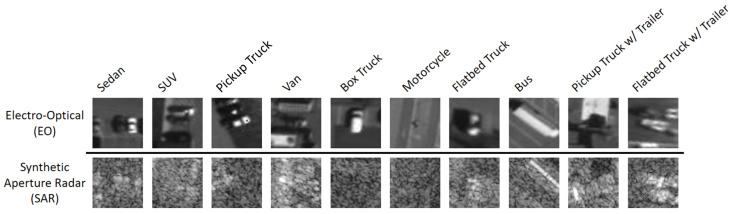
NTIRE 2021 Multimodal Aerial View Object Classification Challenge Dataset [15].

**Figure 6 sensors-24-06054-f006:**
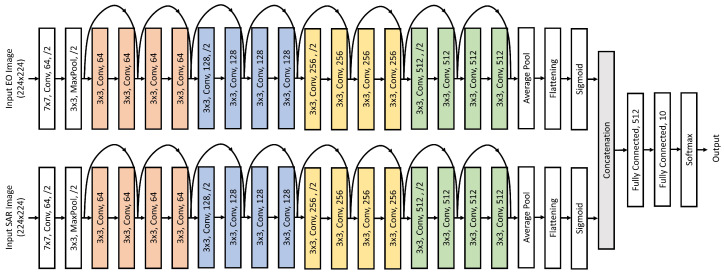
NTIRE 2021 Multimodal Aerial View Object Classification Network with ResNet18 as the backbone.

**Figure 7 sensors-24-06054-f007:**
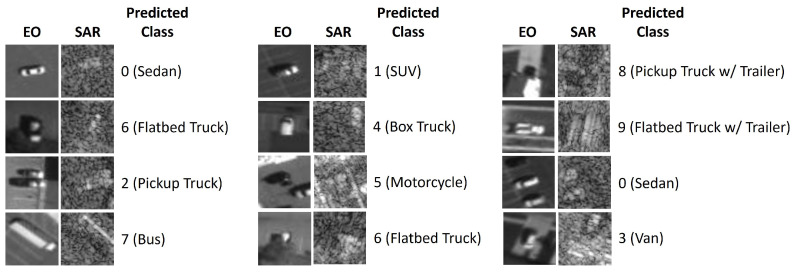
Visualization of NTIRE21 dataset classification using Multimodal Aerial View Object Classification Network.

**Figure 8 sensors-24-06054-f008:**
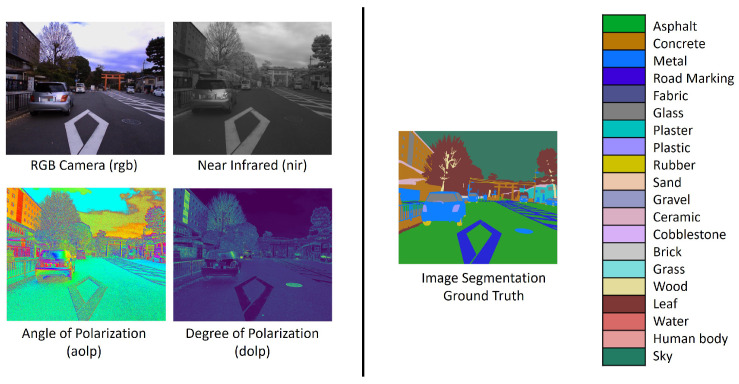
MCubeS Multimodal Material Segmentation Dataset [16].

**Figure 9 sensors-24-06054-f009:**
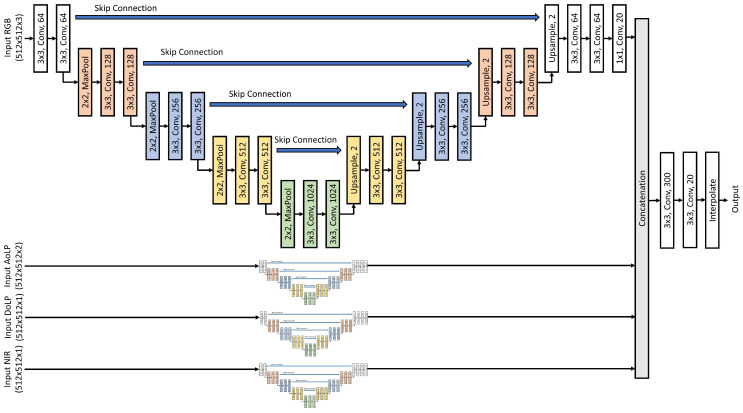
MCubeS Multimodal Material Segmentation Network with UNet as the backbone.

**Figure 10 sensors-24-06054-f010:**
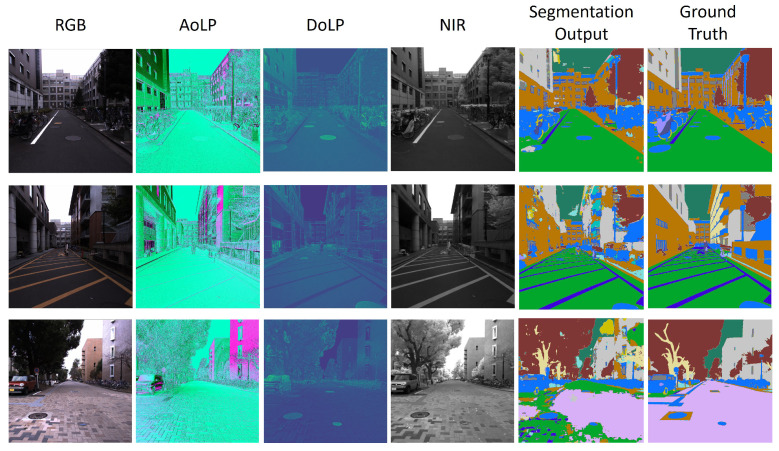
Visualization of MCubeS dataset image segmentation using the Multimodal Material Segmentation Network.

**Figure 11 sensors-24-06054-f011:**
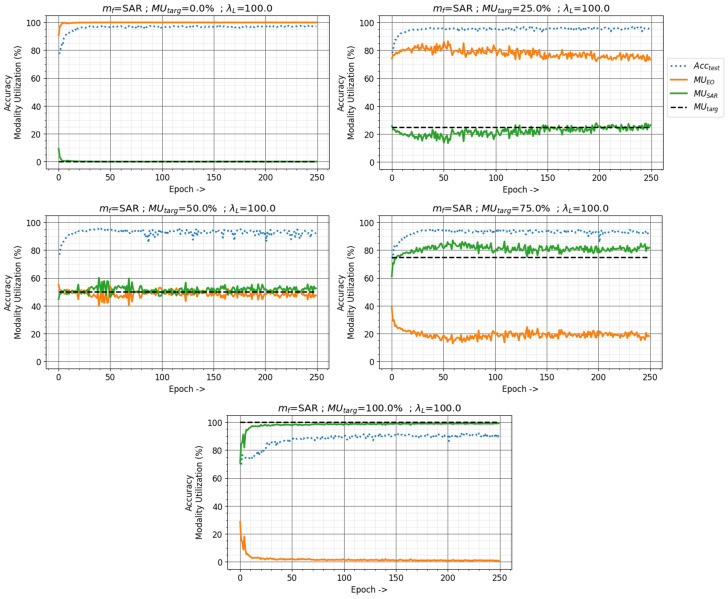
Effects of different target utilization $MU_{target}$ on modality utilization and classification accuracy with modality utilization-based training method in the NTIRE21 dataset. Loss factor $\lambda_{L}=100.0$ with SAR as the focus modality.

**Figure 12 sensors-24-06054-f012:**
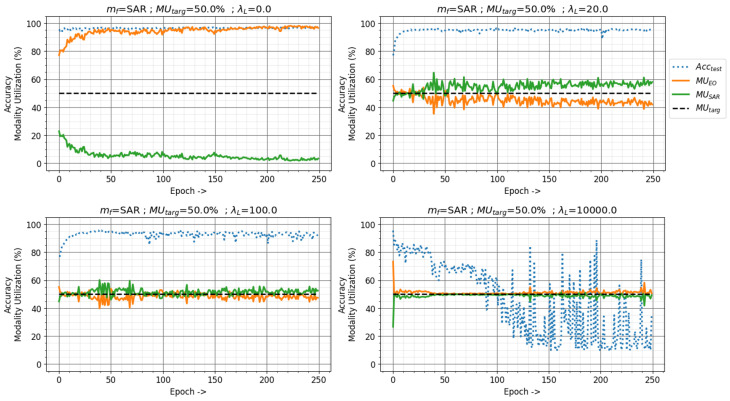
Effects of the loss factor $\lambda_{L}$ on modality utilization and classification accuracy with modality utilization-based training method in NTIRE21 dataset. Target utilization $MU_{target}=50$% with SAR as the focus modality.

**Figure 13 sensors-24-06054-f013:**
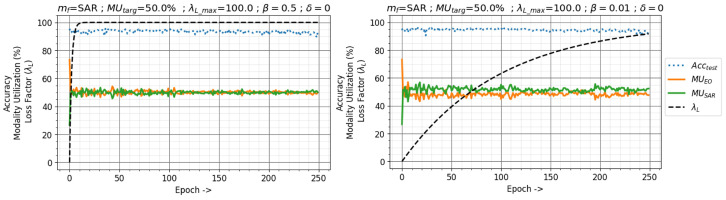
Effects of the loss factor buildup rate β on modality utilization and classification accuracy with modality utilization-based training method in the NTIRE21 dataset. Target utilization $MU_{target}=50$%, Maximum Loss Factor $\lambda_{L\_max}=100$, and buildup delay δ=0 with SAR as the focus modality.

**Figure 14 sensors-24-06054-f014:**
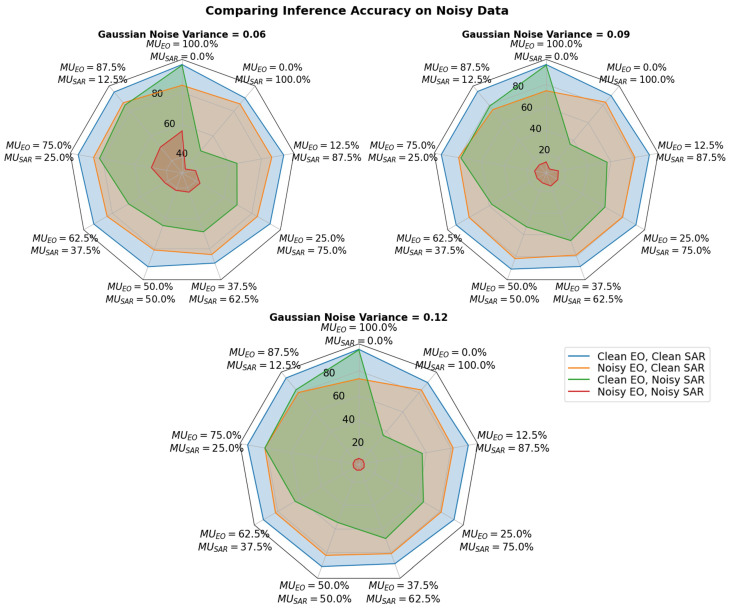
Effects of Gaussian noise with mean = 0 and variance = {0.06,0.09,0.12} in the EO modality, the SAR modality, and both modalities during inference on networks trained with different levels of SAR utilization.

**Table 1 sensors-24-06054-t001:** Performance and modality utilization (MU) for the NTIRE21 and MCubeS datasets [14].

Dataset	Modality	Performance	Modality Utilization (MU) (%)			
		Accuracy (%)	EO		SAR	
NTIRE21	EO	97.5	100.0		-	
	SAR	84.9	-		100.0	
	EO-SAR	97.8	99.59		0.40	
		mIoU	RBG	AoLP	DoLP	NIR
MCubeS	RGB	0.318	100.0	-	-	-
	AoLP	0.266	-	100.0	-	-
	DoLP	0.262	-	-	100.0	-
	NIR	0.270	-	-	-	100.0
	RGB-AoLP-DoLP-NIR	0.374	34.5	19.0	30.9	15.6
	AoLP-DoLP-NIR	0.351	-	67.3	21.0	11.7

**Table 2 sensors-24-06054-t002:** Multimodal fusion network trained on the NTIRE21 dataset for different target utilization $MU_{target}$ with **SAR** as the focus modality. Highest value is represented by a **bold values**.

SAR ${\mathrm{MU}}_{\mathrm{targ}}$ (%)	Acc. (%)	${\mathrm{MU}}_{\mathrm{EO}}$ (%)	${\mathrm{MU}}_{\mathrm{SAR}}$ (%)
0.0	97.1	**99.4**	0.6
12.5	97.6	92.8	7.2
25.0	**97.7**	82.1	17.9
37.5	**97.7**	63.5	36.5
50.0	97.2	47.0	53.0
62.5	96.8	31.7	68.3
75.0	95.3	19.7	80.3
87.5	92.2	13.1	86.9
100.0	84.4	4.8	**95.1**

**Table 3 sensors-24-06054-t003:** Multimodal fusion network trained on NTIRE21 dataset for different target utilization $MU_{target}$ with **EO** as the focus modality. Highest value is represented by a **bold values**.

EO ${\mathrm{MU}}_{\mathrm{targ}}$ (%)	Acc. (%)	${\mathrm{MU}}_{\mathrm{EO}}$ (%)	${\mathrm{MU}}_{\mathrm{SAR}}$ (%)
0.0	84.4	3.9	**96.1**
12.5	92.0	14.3	85.7
25.0	95.3	19.5	80.5
37.5	96.8	30.6	69.4
50.0	97.2	47.0	53.0
62.5	97.6	63.5	36.5
75.0	**97.7**	79.4	20.6
87.5	97.5	94.4	5.6
100.0	97.1	**99.5**	0.5

**Table 4 sensors-24-06054-t004:** Multimodal fusion network trained on MCubeS dataset for different target utilization $MU_{target}$ with loss factor $\lambda_{L}=100.0$ and **RGB** as the focus modality. Highest value is represented by a **bold values**.

RGB ${\mathrm{MU}}_{\mathrm{targ}}$ (%)	mIoU (%)	${\mathrm{MU}}_{\mathrm{RGB}}$ (%)	${\mathrm{MU}}_{\mathrm{AoLP}}$ (%)	${\mathrm{MU}}_{\mathrm{DoLP}}$ (%)	${\mathrm{MU}}_{\mathrm{NIR}}$ (%)
0.0	0.400	1.2	4.2	**54.4**	40.2
12.5	0.403	6.7	0.7	34.2	58.4
25.0	0.388	18.6	10.8	7.3	**63.3**
37.5	0.393	39.4	10.8	16.5	33.3
50.0	0.397	54.9	**15.9**	0.1	29.1
62.5	0.394	68.9	10.5	8.1	12.5
75.0	0.387	84.9	5.4	0.5	9.2
87.5	**0.407**	89.2	5.1	2.6	3.1
100.0	0.403	**96.7**	1.0	0.9	1.4

**Table 5 sensors-24-06054-t005:** Five-fold validation for modality utilization-based training methods **without loss factor warm-up** on NTIRE21 dataset with $m_{f}=SAR$, $\lambda_{L}=100.0$, and $MU_{targ}=0.0$. Instability can be observed in folds 2, 3, and 5. Catastrophic failures in training are represented by red values.

SAR ${\mathrm{MU}}_{\mathrm{targ}}$ (%)	Fold	Acc. (%)	${\mathrm{MU}}_{\mathrm{EO}}$ (%)	${\mathrm{MU}}_{\mathrm{SAR}}$ (%)
0.0	1	97.7	99.9	0.1
0.0	2	44.5	57.8	42.2
0.0	3	51.1	55.5	44.5
0.0	4	98.4	99.9	0.1
0.0	5	48.8	55.9	44.1

**Table 6 sensors-24-06054-t006:** Five-fold validation for modality utilization-based training methods stabilized **with loss factor warm-up** on NTIRE21 dataset with $m_{f}=SAR$, $\lambda_{L}=100.0$, and $MU_{targ}=0.0$.

SAR ${\mathrm{MU}}_{\mathrm{targ}}$ (%)	Fold	Acc. (%)	${\mathrm{MU}}_{\mathrm{EO}}$ (%)	${\mathrm{MU}}_{\mathrm{SAR}}$ (%)
0.0	1	97.7	99.4	0.6
0.0	2	96.9	99.2	0.8
0.0	3	97.6	99.3	0.7
0.0	4	97.2	99.4	0.6
0.0	5	97.0	99.6	0.4

**Table 7 sensors-24-06054-t007:** Effects of Gaussian noise with mean = 0 and variance = {0.06,0.09,0.12} in the **EO modality** (dominant modality) during inference on networks’ trained performance accuracy with different levels of EO utilization.

EO ${\mathrm{MU}}_{\mathrm{targ}}$	No Noise	Noise Var. = 0.06	Noise Var. = 0.09	Noise Var. = 0.12	Average Acc.
100.0	95.1	84.4	74.4	73.7	81.9
87.5	94.6	86.8	74.9	80.3	84.2
75.0	92.8	85.7	80.1	81.4	85.0
62.5	91.8	83.7	80.2	82.1	84.5
50.0	89.8	80.8	82.1	82.3	83.8
37.5	89.1	83.8	78.9	80.9	83.2
25.0	91.2	83.9	79.5	81.1	83.9
12.5	92.2	86.5	81.2	81.7	85.4
0.0	90.3	85.9	83.3	83.0	85.6

## Data Availability

Publicly available datasets were used in this study. NTIRE21 Dataset: Access can be requested from here: https://competitions.codalab.org/competitions/28095 (accessed on 19 May 2022); MCubeS Dataset: Can be accessed from here: https://github.com/kyotovision-public/multimodal-material-segmentation (accessed on 19 September 2022).

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
