# Peer review of "Regulating Modality Utilization within Multimodal Fusion Networks"

_sensors, 2024, doi:10.3390/s24186054_

Round 1
Reviewer 1 Report
Comments and Suggestions for Authors
This paper proposes a modality utilization-based training method for multimodal fusion. Overall, this article provides a relatively strong theoretical discussion. However, there remains the following issues to be addressed in the paper:
1. Before the decision layer, the multimodal features are fused through a simple concatenation operator. Is it possible to replace the concatenation operator with convolutional or transformer networks to obtain better performance?
2. Please introduce the architecture of the decision layers.
3. The MCubeS dataset used in the experiments appears not match the aerial images mentioned in the title.
4. In many scenarios, SAR shows clear advantages over electro-optical. However, as shown in the Table 1, there is a huge disparity in the percentage of modality utilization on the NTIRE21 dataset. Whether the problem that cause this phenomenon is that the dataset is not suitable for validating the proposed method?
5.Please compare the proposed model with more SOTA methods.
Comments on the Quality of English LanguageMinor editing of English language required.
Author Response
Comment 1 Before the decision layer, the multimodal features are fused through a simple concatenation operator. Is it possible to replace the concatenation operator with convolutional or transformer networks to obtain better performance?
Thank you for the insightful suggestion. Indeed, we consider it possible to attain performance gains by means of more advanced fusion methods, such as convolutional or transformer networks –even though our current simple concatenation already achieves very high accuracy levels at 97%. However, because the primary focus of our paper is to explore the regulation of modality utilization and its impact on robustness against noise, we opted to avoid delving into fusion optimization and maintain a simple fusion operator (concatenation), so that we can more clearly assess the effects of our proposed contribution.
Comment 2 Please introduce the architecture of the decision layers.
Great suggestion. We have incorporated additional details about the architecture of the decision layers in Sections 4.1.1 and 4.1.2. This includes network architecture figures to provide a clearer understanding of the structure and functionality of the decision layers. Please refer to the adde Figures 6 & 7 on page 10 and Figures 9 & 10 on page 11.
Comment 3 The MCubeS dataset used in the experiments appears not match the aerial images mentioned in the title.
Thank you for your insightful comment. You are correct that the presented method is not limited to aerial imagery classification and can indeed be applied to various domains and machine learning problems. While our primary results and analysis focus on an aerial imagery dataset, we have also demonstrated the versatility of our method by experimenting with the MCubeS image segmentation dataset. This dataset was previously used in our paper introducing the modality utilization metric (https://ieeexplore.ieee.org/document/10195122). To best address your helpful comment and reflect the broad applicability of our method, we have revised the title to "Regulating Modality Utilization within Multimodal Fusion Networks."
Comment 4 In many scenarios, SAR shows clear advantages over electro-optical. However, as shown in the Table 1, there is a huge disparity in the percentage of modality utilization on the NTIRE21 dataset. Whether the problem that cause this phenomenon is that the dataset is not suitable for validating the proposed method?
We appreciate your observation regarding the disparity in modality utilization on the NTIRE21 dataset. As highlighted in Table 1, electro-optical (EO) imagery indeed dominates the dataset, which primarily excludes scenarios with cloudy skies and night-time conditions where synthetic-aperture radar (SAR) might demonstrate its advantages over EO. This dataset's limitations in capturing diverse environmental conditions are well-documented (as cited: https://openaccess.thecvf.com/content/CVPR2021W/NTIRE/html/Liu_NTIRE_2021_Multi-Modal_Aerial_View_Object_Classification_Challenge_CVPRW_2021_paper.html). Our method's performance, despite the dataset bias towards EO, underscores its significance in mitigating the inherent bias towards a singular modality. By demonstrating competitive performance under such constraints, our approach contributes towards broadening the applicability and robustness of multi-modal object classification methods in real-world scenarios.
Comment 5 Please compare the proposed model with more SOTA methods.
Thank you for your valuable suggestion. We appreciate the importance of comparing our model with more state-of-the-art methods. To our knowledge, this work is the first to explicitly define the reliance of the network on different modalities during training. While we found several relevant works, they focus on different aspects and are not directly comparable to our approach. For instance, some methods regulate the learning rate of each modality branch, introduce regularization terms to reduce bias among modalities, or remove modalities with low utility. Although these techniques may improve network performance, they do not examine or pursue a balanced reliance on each modality. Additionally, our work emphasizes robustness against noisy modalities rather than performance in ideal, noise-free conditions, making direct comparisons with these methods less relevant. However, to best address your comment, we have added the needed citations to cover the latest research in multimodal fusion. Please refer to lines 107-153 on page 3-4. Due to the limited timeframe of the review process and the extensive duration new experiments would require, we regrettably could not conduct additional comparisons at this time –and frame them in a coherent context with the proposed approach. We hope that you will find our paper appropriately focused on our main scope and look forward to exploring these comparisons in future work.
Reviewer 2 Report
Comments and Suggestions for Authors
The dataset needs to be described more.
Results need to be quantified in the abstract as well as in the conclusion section.
The research questions part need to be written in a paragraph.
It is better to abbreviate everything in the text rather than after the manuscript.
Comments on the Quality of English LanguageThe quality of English language is acceptable.
Author Response
Comment 1 The dataset needs to be described more.
Thank you for your suggestion. We have addressed this by adding more detailed information about the dataset in Sections 4.1.1 and 4.1.2. This includes comprehensive descriptions and figures illustrating the network architecture and data characteristics. Please refer to the adde Figures 6 & 7 on page 10 and Figures 9 & 10 on page 11.
Comment 2 Results need to be quantified in the abstract as well as in the conclusion section.
Thank you for your valuable suggestion. We have addressed this by adding quantified results to both the abstract and conclusion sections. These additions highlight our method's ability to maintain target modality utilization and the improvements in noise robustness, providing clearer insights into our findings. Please see the updated abstract (lines 7-17) and conclusion section (lines 602-612). We also added information to back numbers these up, see lines 386-388 on page 12, lines 392-394 on page 13, lines 420-426 on page 15, and lines 527-529 on page 18.
Comment 3 The research questions part need to be written in a paragraph.
Thank you for your insightful comment. We have addressed this by incorporating the research questions into a cohesive paragraph within the introduction section. This integration helps to clearly outline the primary objectives and scope of our study. Please see the updated paragraph in lines 65-79 on page 2.
Comment 4 It is better to abbreviate everything in the text rather than after the manuscript.
Thank you for your feedback. We have ensured that all abbreviations are introduced and defined within the text. Additionally, we have retained the abbreviations list at the end of the manuscript, as it is part of the MDPI Sensors template requirements.
Reviewer 3 Report
Comments and Suggestions for Authors
This manuscript proposes a novel modality utilization-based training method for multimodal fusion networks. It addresses the bias caused by the overutilization of a single modality, thereby enhancing the potential benefits of multimodal data. The proposed method has been validated in image classification and segmentation tasks. While the manuscript holds research value, there are several shortcomings that need to be addressed. If the following issues are revised and improved, I will consider recommending the manuscript for publication.
- The core content of this manuscript focuses on the fusion and application of multimodal data in aerial imagery. However, the Introduction section scarcely mentions the current state of research and existing problems in the field of multimodal fusion for aerial applications. Additionally, the Introduction lacks a discussion of recent domestic and international research progress, particularly studies on multimodal fusion from 2023 onwards.
- In Figures 1 and 2, the role of the loss function L is not indicated. Please specify its position in the figures.
- The research questions presented in Section 3.4 are better suited for the Introduction, allowing readers to directly grasp the issues the manuscript aims to address.
- Sections 4.1.1 and 4.1.2 lack diagrams of the classification and segmentation network frameworks. It is recommended to provide schematic diagrams of the multimodal fusion network framework. Without these, it is difficult to understand where the proposed Modality Utilization-based Training method is applied in different tasks. For instance, convert the network structures described in lines 269-273 and lines 294-297 into diagrams.
- The software and hardware configurations used in the experiments are not detailed. It is advisable to describe them comprehensively.
- In lines 278-279 and lines 299-300, the experiments are conducted using a limited number of samples for training and testing. Have overfitting issues been considered?
- The accuracies (Acc) presented in Tables 2 and 3 show minimal differences between SAR MUtarg and EO MUtarg. This suggests that the small training sample size may have led to overfitting. It is recommended to increase the number of training samples and then recalculate the accuracies. This applies to the segmentation tasks as well.
- In Tables 2, 3, and 4, it is suggested to highlight the best results, such as using bold font.
- Sections 4.1.1 and 4.1.2 should include visualizations of the predicted classification and segmentation results. Currently, only quantitative results are presented, and there is a lack of visual and intuitive demonstrations of the outcomes.
Need to fine-tune English quality.
Author Response
Comment 1 The core content of this manuscript focuses on the fusion and application of multimodal data in aerial imagery. However, the Introduction section scarcely mentions the current state of research and existing problems in the field of multimodal fusion for aerial applications. Additionally, the Introduction lacks a discussion of recent domestic and international research progress, particularly studies on multimodal fusion from 2023 onwards.
Thank you for your insightful feedback. We acknowledge the importance of providing a comprehensive overview of the current state of research and existing challenges in multimodal fusion for aerial applications within the Introduction section. While our manuscript includes a dedicated section for related work following the introduction that covers multimodal aerial imagery and associated challenges, we understand the need for a more robust introduction to set the context effectively. To address this comment, we have enriched the Introduction section by adding more recent citations that highlight advancements in multimodal fusion from 2023 onwards. Please see page 2 lines 39-43, lines 53-62. Additionally, we have expanded on the general challenges encountered in the field of aerial imagery. Please refer to lines 107-153 on page 3-4.
Comment 2 In Figures 1 and 2, the role of the loss function L is not indicated. Please specify its position in the figures.
Thank you for your feedback. In Figure 1, the standard loss function (e.g., Cross Entropy loss) is implicitly understood within the context of neural network training for the modality utilization metric, hence we did not explicitly indicate its position. However, we agree that specifying the position of the modified loss function in Figure 2, which illustrates the modality utilization-based training method, is crucial for clarity. We have updated Figure 2 on page 6 accordingly. These updates aim to enhance the clarity and comprehensibility of our figures, addressing your suggestion.
Comment 3 The research questions presented in Section 3.4 are better suited for the Introduction, allowing readers to directly grasp the issues the manuscript aims to address.
Thank you for your suggestion. We have integrated the research questions into the introduction as advised. Please see the updated paragraph in lines 65-79 on page 2.
Comment 4 Sections 4.1.1 and 4.1.2 lack diagrams of the classification and segmentation network frameworks. It is recommended to provide schematic diagrams of the multimodal fusion network framework. Without these, it is difficult to understand where the proposed Modality Utilization-based Training method is applied in different tasks. For instance, convert the network structures described in lines 269-273 and lines 294-297 into diagrams.
Thank you for the suggestion. We have now included schematic diagrams of the classification and segmentation network frameworks in Sections 4.1.1 and 4.1.2, illustrating the proposed multimodal fusion network framework. These diagrams clarify the application of our Modality Utilization-based Training method across various tasks. Please refer to the adde Figures 6 & 7 on page 10 and Figures 9 & 10 on page 11.
Comment 5 The software and hardware configurations used in the experiments are not detailed. It is advisable to describe them comprehensively.
Thank you for the suggestion. We have now provided comprehensive details of the hardware and software configurations used for all experiments in Section 5.1. Please see lines 358-363 on page 12.
Comment 6 In lines 278-279 and lines 299-300, the experiments are conducted using a limited number of samples for training and testing. Have overfitting issues been considered?
Thank you for your comment. We have addressed potential overfitting concerns in our experiments by employing 5-fold cross-validation to ensure consistent network performance across multiple datasets. Additionally, we monitored loss curves to verify absence of overfitting. Please refer to the attached examples for more details.

Additionally, we have updated the paper to include details on saving models with the best performance based on lowest MU loss, and using early stopping techniques to further mitigate overfitting risks. Please refer to lines 325-327 on page 10 and lines 350-352 on page 12. For the MCubeS dataset, please also note that no samples have been removed from the dataset; the original dataset is made up of 500 samples in total (Link to dataset: https://github.com/kyotovision-public/multimodal-material-segmentation).
Comment 7 The accuracies (Acc) presented in Tables 2 and 3 show minimal differences between SAR MUtarg and EO MUtarg. This suggests that the small training sample size may have led to overfitting. It is recommended to increase the number of training samples and then recalculate the accuracies. This applies to the segmentation tasks as well.
Thank you for your feedback. We have carefully considered your observation regarding the minimal differences in accuracies between SAR MU_targ and EO MU_targ as presented in Tables 2 and 3. The presence of redundant information between the two modalities is a common observation in such multimodal datasets. In our previous work, we discussed the redundancy present in different modalities (https://ieeexplore.ieee.org/document/10195122). Redundant information allows for the reduction in the utilization of one modality up to the point where the unique information in that modality begins to be compromised. To further clarify these observations, we have expanded on this point in Section 5.1, discussing how the redundancy impacts the interpretation of data in Tables 2 and 3. Please refer to lines 394-399 on page 13-14.
Comment 8 In Tables 2, 3, and 4, it is suggested to highlight the best results, such as using bold font.
Thank you for your suggestion to highlight the best results in Tables 2, 3, and 4. We have implemented your recommendation by bolding the highest value in each column of these tables. This includes highlighting the highest Acc./mIoU and emphasizing the case with high utilization of that modality. Please refer to Table 2 & 3 on page 13 and Table 4 on page 14.
Comment 9 Sections 4.1.1 and 4.1.2 should include visualizations of the predicted classification and segmentation results. Currently, only quantitative results are presented, and there is a lack of visual and intuitive demonstrations of the outcomes.
Thank you for your comment. We have now included visualizations in Sections 4.1.1 and 4.1.2 to provide intuitive demonstrations of the predicted classification and segmentation results alongside the quantitative results. Please refer to Figure 7 on page 10 and Figure 10 on page 11.
Round 2
Reviewer 1 Report
Comments and Suggestions for Authors
The revised paper resolved most of my previous concerns. I think the current manuscript achieves the standard to be accepted for publication.
Comments on the Quality of English LanguageMinor editing of English language required.